# Mapping Geographical Differences and Examining the Determinants of Childhood Stunting in Ethiopia: A Bayesian Geostatistical Analysis

**DOI:** 10.3390/nu13062104

**Published:** 2021-06-19

**Authors:** Kedir Y. Ahmed, Kingsley E. Agho, Andrew Page, Amit Arora, Felix Akpojene Ogbo

**Affiliations:** 1Translational Health Research Institute, School of Medicine, Western Sydney University, Locked Bag 1797, Penrith, NSW 2571, Australia; K.agho@westernsydney.edu.au (K.E.A.); a.page@westernsydney.edu.au (A.P.); a.arora@westernsydney.edu.au (A.A.); f.ogbo@westernsydney.edu.au (F.A.O.); 2College of Medicine and Health Sciences, Samara University, Samara 132, Ethiopia; 3School of Health Sciences, Western Sydney University, Locked Bag 1797, Penrith, NSW 2751, Australia; 4African Vision Research Institute (AVRI), University of KwaZulu-Natal, Durban 4041, South Africa; 5Health Equity Laboratory, Campbelltown, NSW 2560, Australia; 6Oral Health Services, Sydney Local Health District and Sydney Dental Hospital, NSW Health, Surry Hills, NSW 2010, Australia; 7Discipline of Child and Adolescent Health, Faculty of Medicine and Health, Sydney Medical School, The University of Sydney, Westmead, NSW 2145, Australia; 8Barmera Medical Clinic, Lake Bonney Private Medical Clinic, 24 Hawdon Street, Barmera, SA 5345, Australia

**Keywords:** undernutrition, stunting, geo-statistics, inequality, Ethiopia, children

## Abstract

Understanding the specific geographical distribution of stunting is essential for planning and implementing targeted public health interventions in high-burdened countries. This study investigated geographical variations in the prevalence of stunting sub-nationally, and the determinants of stunting among children under 5 years of age in Ethiopia. We used the 2016 Ethiopia Demographic and Health Survey (EDHS) dataset for children aged 0–59 months with valid anthropometric measurements and geographic coordinates (*n* = 9089). We modelled the prevalence of stunting and its determinants using Bayesian geospatially explicit regression models. The prevalence of stunting among children under five years was 36.3% (95% credible interval (CrI); 22.6%, 51.4%) in Ethiopia, with wide variations sub-nationally and by age group. The prevalence of childhood stunting ranged from 56.6% (37.4–74.6%) in the Mekelle Special zone of the Tigray region to 25.5% (10.5–48.9%) in the Sheka zone of the Southern Nations, Nationalities and Peoples region. Factors associated with a reduced likelihood of stunting in Ethiopia included non-receipt of breastmilk, mother’s BMI (overweight/obese), employment status (employed), and higher household wealth, while the enablers were residence in the “arid” geographic areas, small birth size of the child, and mother’s BMI (underweight). The prevalence and determinants of stunting varied across Ethiopia. Efforts to reduce the burden of childhood stunting should consider geographical heterogeneity and modifiable risk factors.

## 1. Introduction

The first 2000 days of life (from conception to age 5 years) provide a great window of opportunity for improving child survival, good health and development, and these benefits extend across the life course [1,2,3,4]. In this age group, appropriate nutrition, psychosocial interactions and a built environment are essential to meet childhood developmental and nutritional requirements. However, early years nutritional deficiencies (that is, becoming underweight, stunted or wasted) are associated with short- and long-term adverse consequences among children [1,2,5,6,7,8]. Suppressed immunity, increased risk of morbidity and mortality, and lower school performance have been reported in children with stunting [1,2,5,6,7,8]. Childhood stunting is one of the strongest indicators for assessing the overall health and well-being of children [9].

Ethiopia is the second most populous country in Africa (after Nigeria), with over 112 million people [10] and has the fastest growing economy in the continent [11]. Evidence from the Ethiopia Demographic and Health Survey (EDHS) reports showed a steady decrease in the prevalence of stunting from 58% in 2000 to 37% in 2019 [12,13,14]; however a significant number of children are stunted. The reduction in childhood stunting is likely due to ongoing national and subnational efforts to reduce childhood malnutrition, including the “Seqota” Declaration to end childhood malnutrition by 2030 [15] and the National Nutrition Program to end hunger by 2030 [16]. While socioeconomic improvements have been reported in Ethiopia, about 23% of the population is still socio-economically disadvantaged, and more than 5 million children under five are reported to be stunted [17]. Additionally, one in 15 Ethiopian children dies before the age of five years [17], and undernutrition accounts for an estimated 30% of these deaths [18]. 

In Ethiopia, more than 40% of the population resides in arid and semi-arid areas [19], where agricultural production is lower. The country also experiences frequent natural and manmade disasters, including droughts, flooding, rising temperature, and internal conflicts [20,21], These events increase the vulnerability to low-yield agricultural production that can subsequently lead to food insecurity [22,23] and childhood undernutrition [24,25,26].

Several studies conducted in Ethiopia have shown that suboptimal infants and young child feeding (IYCF) [27,28,29,30], food insecurity [31,32], lower maternal education [27,28,29,30,33], maternal underweight [33,34], childhood infections (e.g., diarrhea and pneumonia) [27,29] were associated with childhood stunting. Other relevant factors associated with stunting included poverty [29], unsafe hygiene and sanitation [29], climate change, low crop production, the higher market price of food, and natural and manmade disasters [35]. While previous studies in Ethiopia provide valuable information, there are gaps in knowledge relating to the burden of stunting. Firstly, some of the studies (e.g., Haile et al., 2016) used older EDHS datasets, which does not reflect the current socio-economic, health, geographic, political and in-country migration status in Ethiopia. Secondly, these past studies did not examine the subnational-level prevalence of stunting to facilitate within-country comparisons, as national-level information can mask important geographic differences. Thirdly, previous studies did not examine the associations between geo-located and/or geo-referenced determinants (i.e., environmental and climatic factors) of childhood stunting at the national level in Ethiopia. Finally, the assessment of the geographical variability of the prevalence of stunting in lower administrative levels can provide locally relevant public health information that will help decision-makers and efficient resource allocation in nutritional interventions for Ethiopian children most in need.

The present study aims to: (i) examine the spatial variability of stunting prevalence among children under 5 years of age at the subnational level in Ethiopia by age groups (0–23 months and 24–59 months, the child age subcategorization is essential for the design and implementation of targeted childhood nutrition programs to improve efficacy and effectiveness) [9]; and (ii) investigate the proximal and contextual factors associated with stunting among children under 5 years of age at the national level in Ethiopia. 

## 2. Materials and Methods 

### 2.1. Data Sources

Data were based on the nationally representative 2016 EDHS (*n* = 9089). The survey was implemented by the Central Statistical Agency (CSA) and Inner City Fund (ICF) International and funded by the United States Agency for International Development [36], and the Government of Ethiopia [13,17,37,38]. The 2016 EDHS used a two-stage stratified cluster sampling technique to select the study participants. In stage one, 645 enumeration areas (EAs) were randomly selected in each sampling stratum with probability proportional to EA size, using the 2007 Ethiopia Population and Housing Census [39]. A complete household listing was conducted to develop a sampling frame for the selection of households. In stage two, a systematic random sampling technique was used to select a fixed number of 28 households in each EA. Out of 16,583 eligible women of reproductive age (15–49 years of age) from the selected households, 15,683 were successfully interviewed, yielding a response rate of 94.6%. 

The 2016 EDHS collected information on maternal and child health indicators, including height and weight measurements for children under five children. A total of 10,752 children under five were sampled from 645 clusters in the 2016 EDHS. Our study included 9089 children who resided in 622 clusters, where valid geographic coordinates and anthropometric data are collected. The detailed methodology for the 2016 EDHS survey is reported elsewhere [17]. The 2016 EDHS collected geographic coordinates for each cluster using the Global Positioning System (GPS) receivers. To keep the confidentiality of respondents in each cluster, GPS coordinates were displaced (geo-masking) by up to 10 km for rural clusters and 2 km in urban clusters [40]. For surveys collected after 2008, the GPS displacement is further restricted to the second administrative level units (“Zones” in the Ethiopian context) [41,42]. The 2016 EDHS clusters where the prevalence of stunting were calculated are presented in Appendix A.

For each georeferenced EDHS cluster, climatic and demographic data were extracted from publicly available remote sensing raster and vector data sources. The raster data (images and modelled surfaces) rely on pixels or cells to convey their data values, while vector data (points, lines, and polygons) depend on the discrete location or boundary of a feature [40]. During the extraction of the geo-covariates, the EDHS circularly buffered the data within 2 km for urban points and 10 km for rural points to ensure all points (including coordinates with the maximum displacement) fell within the radius of the circular buffer and to account for the variation in pixel size in the data sources [40,41]. The detailed procedure on the extraction of geo-covariates was published in the DHS manual [40].

### 2.2. Outcome Variable

The main outcome variable for this study was stunting, measured using the World Health Organization (WHO) Child Growth Standards of height-for-age z-scores (HAZ) [43]. The length or height was measured using Shorr measuring board [44]. Stunting was measured using the nutritional index of HAZ that was calculated in standard deviation (SD) units from the median of the WHO reference population for age. Children were classified as stunted if < −2.0 SD HAZ-score, consistent with the 2016 EDHS report [17] and previously published studies [45,46]. For this study, the nutritional status of children was classified on a dichotomous scale (“1 = Yes/stunted” or “0 = No/not stunted”), consistent with EDHS reports and previously published studies [45,46].

### 2.3. Study Variables

We adapted the United Nation’s Children Fund (UNICEF) [7] and the WHO [47] conceptual frameworks for undernutrition [47] and used them in past studies from low- and middle-income countries (LMICs) [46,48,49] (Figure 1). The study variables were broadly classified into putative proximal and contextual factors. The proximal factors have immediate biological (e.g., not eating enough or eating that lack growth-promoting nutrients) and pathophysiological (infections or diseases that can cause poor nutrient intake, absorption or utilization) relationships with stunting [7,47,49]. The contextual factors selected are based on associations with the familial, societal and community contexts where the child resided [7,47,49].

The proximal factors included IYCF (early initiation of breastfeeding, minimum dietary diversity, minimum meal frequency, type of food in the past 24 h, duration of breastfeeding, and bottle feeding) and child morbidity (diarrhea, acute respiratory infection (ARI) and anemia). Proximal contextual factors included maternal factors (mother’s nutritional status, mother’s and father’s education, mother’s and father’s employment and mother’s age), household factors (household wealth index, source of drinking water, type of toilet system and cooking fuel), health service factors (frequency of antenatal care [ANC] visits and place of birth), and child factors (perceived birth size, child age, and birth order). Environmental contextual factors included media exposure (listening to the radio, reading a magazine, and watching television), and climatic factors (daytime land surface temperature (DLST), annual rainfall, aridity and number of wet days per year). The climatic factors were selected based on past studies and their influence on poverty and crop production, and with consequent effect on childhood stunting [49,50,51,52]. Appendix A provides detailed information on the definitions, classifications and data sources of the selected study variables.

### 2.4. Analytical Strategy

Frequencies and percentages of proximal and contextual factors were initially calculated. This was followed by a descriptive analysis that estimated the prevalence of stunting according to age groups (0–23 months and 24–59 months) by each study factor. This age group classification was used to examine associations between age-limited study factors (e.g., IYCF used for children 0–23 months of age) with stunting, and to calculate geographical variations in stunting across each age group (0–23 months and 24–59 months). In addition, interaction checks of regression models showed significant differences in the measures of associations for independent variables (wealth, baby size at birth, anemia, maternal education, and type of toilet system) across the age classification (0–23 months and 24–59 months) (Table 1).

All descriptive analyses including frequencies and percentages were calculated using the “svydesign” function from the “survey” package to adjust for sampling weights, clustering and stratification in R (R Core Team, Austria) [53]. Children aged 0–23 with mother’s perceived small birth size had a higher prevalence of stunting compared to larger than average birth size (33.0% vs. 24.6%). A higher prevalence of stunting was found among children aged 0–59 months whose mothers did not have schooling compared to those with secondary or higher education (41.5% vs. 19.8%). Additional information on the prevalence of stunting over proximal and contextual factors is presented in Appendix A.

Bayesian geostatistical models were used to examine associations between proximal and contextual factors with stunting by age group, while accounting for the geographical dependence of EDHS clusters, consistent with previous studies [54,55,56,57,58]. Bayesian geostatistical models are models of point-referenced data that include a spatially structured random effect implemented with a Bayesian method of inference framework [59]. The geographical dependence of clusters was incorporated into the models as spatially correlated higher level random effects by assuming that the spatial autocorrelation decays when the distance between locations increases [60]. The Bayesian geostatistical models were also used to produce second administrative-level prevalence of stunting and spatially explicit maps over the different administrative levels of Ethiopia. All geostatistical models were fitted using the Bayesian framework for estimating the posterior distribution of fixed effects (such as odds ratios (ORs), prevalence, standard deviations, and 95% credible intervals (CrIs)) and random parameters (such as kappa, variances and ranges). Appendix A is presented to show the presence of global autocorrelation using Moran’s I.

The Bayesian geostatistical models specified were conducted in five stages. In stage one, a gridded data with geo-covariates and household survey data with geo-coordinates from the EDHS were imported to the R environment for geostatistical computing [53]. In R, the imported data (i.e., geo-covariates and household survey data with geo-coordinates) were merged using “cluster-id” as a unique identifier. In stage two, models with improved possible combinations of study variables were fitted for variable selection using Watanabe–Akaike information criterion (WAIC) and deviance information criterion (DIC) [61,62], consistent with the past studies [61]. At this stage, the preliminary model selection using WAIC and DIC removed study variables such as type of cooking fuel, maternal age, delivery assistance, vaccination status, birth order, birth interval., media exposures (i.e., radio, television and magazine), and climatic factors (e.g., proximity to water bodies and enhanced vegetation index). In stage three, the Stochastic Partial Differential Equation (SPDE) that assumed a stationary and isotropic Matérn covariance matrix was used to specify the spatial data process, and to calculate the spatial autocorrelation structure of the study region using an artificial set of vertices called a mesh (Appendix A). Unlike areal geospatial data, the point referenced data do not have explicit neighbors to calculate the spatial autocorrelation, and thus we artificially created mesh to represent the neighboring structure of the study region. Subsequently, a projector matrix was created to link the observed locations (EDHS clusters) with the created mesh vertices (that were weighted based on their distance from the observed locations) to serve as explicit neighbors.

In stage four, non-spatial and geospatial grouped binomial regression models were fitted using ‘logit’ link function by calculating cluster level proportion of stunting (using the number of stunted children as numerator and the total number of children as a denominator) to examine associations between proximal and contextual factors and stunting. The non-spatial models were fitted to examine the improvements in model variance after considering the spatial autocorrelation as a random effect in the geostatistical models. The Bayesian geostatistical models were fitted to examine associations between the study variables and stunting. To predict the prevalence of stunting at high resolution grids (un-sampled locations), only proximal and contextual factors with available raster surfaces (such as maternal education, antenatal care visits 4+, place of birth, type of toilet system, source of drinking water, aridity, number of wet days, DLST, and annual average rainfall) were accounted in final prediction models [59]. Maps and second administrative level (referred to as “Zonal level” in Ethiopia) prevalence of stunting were estimated and reported using the output from these geostatistical prediction models. In stage five, all models were grouped using a random selection process into a “training set” (75% of the sample) and a “test set” (remaining 25% of clusters), consistent with previously published studies [61,63]. Detailed information on the model formulation, development and implementation is provided as File S1.

The Integrated Nested Laplace Approximation (INLA) algorithm was used to conduct all models using the R-INLA package [64]. Bayesian inference using INLA is a computationally less intensive alternative to the Markov Chain Monte Carlo (MCMC) that is designed to approximate the MCMC estimations, particularly in latent Gaussian models such as generalized linear mixed models, and spatial and spatio-temporal models [64,65]. Gridded predicted risk maps at un-sampled locations were produced on a regular grid of 112,346 pixels on 5 km by 5 km spatial resolution covering all of Ethiopia. All Bayesian inferences used non-informative priors in the estimation of posterior parameters, including ORs, 95% CrIs, ranges and variances. Non-informative priors with normal distributions of mean and precision *n* (0, 0/τ, τ = 0) for intercepts, and mean and precision *n* (0, 0.001) for regression coefficients were used. For random effects, default priors of gamma distributions with gamma (1, 0.00005) for spatial decays and inverse gamma priors for variance were specified. ORs with 95% CrIs were estimated and reported as the measure of association between proximal and contextual factors and stunting in this study. A CrI is an interval in which an unobserved parameter value falls with a given probability. It is the Bayesian equivalent of the confidence interval; however, unlike a confidence interval, it is dependent on the prior distribution, specific to the situation [34].

### 2.5. Ethics

The survey was conducted after ethical approval was obtained from the National Research Ethics Review Committee (NRERC) in Ethiopia. During the survey, permission from administrative offices and verbal consent from study participants was obtained before the commencement of data collection. For this study, the dataset was obtained from Measure DHS/ICF with permission.

## 3. Results

### 3.1. Geographical Patterns of Stunting and Notable Subnational Variations

The prevalence of stunting among children under-five years was 36.3% (95% credible interval (CrI); 22.6%, 51.4%) at the national level in Ethiopia, with wide variations at the subnational level and by age group. The prevalence of stunting among children aged 0–23 months ranged from 19.1% to 47.7% with a median of 31.2%. Among children aged 24–59 months, the prevalence of stunting ranged from 24.9% to 63.8% with a median of 46.2% (Figure 2, Figure 3, Figure 4 and Figure 5).

In children aged 0–23 months, the prevalence of stunting was higher in the South-East zone (*p* = 37.4%; 95% CrI: 18.3, 61.9) and the Mekelle Special zone (*p* = 36.8%; 95% CrI: 18.6%, 59.6%) of the Tigray region, while the lowest prevalence was reported in the Sheka zone of the SNNP region (*p* = 25.5%; 95% CrI: 10.5%, 48.9%) [Appendix A]. Children aged 24–59 months who resided in the Mekelle Special zone of the Tigray region (*p* = 56.6%; 95% CrI: 37.4, 74.6) and the Oromia Special zone of the Amhara region (*p* = 55.3%; 95% CrI: 32.7, 76.1) had a higher prevalence of stunting. Children who were from the Etang Special zone of the SNNP region had the lowest prevalence of stunting among children aged 24–59 months (*p* = 29.4%; 95% CrI: 13.8%, 52.0%) [Appendix A].

### 3.2. Factors Associated with Stunting among Children 0–23 Months of Age

In the geospatial regression model, which accounted for the spatial autocorrelation structure, children who were breastfed for more than 12 months (odds ratio (OR) = 2.03; 95% CrI: 1.36, 3.05) and those whose mothers were underweight (OR = 1.36; 95% CrI: 1.11, 1.65) were more likely to be stunted compared to their counterparts. In the same model, children who were perceived to be smaller than average (OR = 1.35; 95% CrI: 1.08, 1.60) and those who resided in the “arid” geographic areas (OR = 2.21; 95% CrI: 1.22, 4.02) were more likely to be stunted compared to their counterparts. Maintaining the influence of spatial autocorrelation and other covariates constant, children who did not receive breastmilk within 24 hours prior to the survey (OR = 0.59; 95% CrI: 0.38, 0.89), and those with overweight/obese mothers (OR = 0.47; 95% CrI: 0.31, 0.69) were less likely to be stunted compared to their counterparts. Children from middle-income households (OR = 0.68; 95% CrI: 0.53, 0.89), and those with employed mothers (OR = 0.68; 95% CrI: 0.51, 0.91) had a lower odds of stunting, following the influence of spatial autocorrelation accounted (Table 2).

In children aged 0–23 months, the spatial range, where the spatial autocorrelation became negligible (less than 0.1), was 52.2 km (95% CrI: 15.5, 98.8), and the spatial variance was 0.24 (95% CrI: 0.10, 0.39) (Table 2).

### 3.3. Factors Associated with Stunting among Children 24–59 Months of Age

In the geospatial regression model, children who did not receive breastmilk (OR = 0.57; 95% CrI: 0.48, 0.67) and those who resided in rich-income households (OR = 0.70; 95% CrI: 0.58, 0.85) had lower odds of being stunted compared to those who were exclusively breastfed and those from poor income households, respectively. In the same model, children who were anemic (OR = 1.73; 95% CrI: 1.52, 1.96), and those who were perceived to be smaller than average (OR = 1.64; 95% CrI: 1.39, 1.92) were associated with higher odds of stunting compared to their counterparts. After accounting for the spatial autocorrelation, children who resided in the “arid” geographic locations were more likely to be stunted compared to those who resided in the “wet” geographic locations (OR = 2.02; 95% CrI: 1.11, 3.65) (Table 2).

In children aged 24–59 months, the spatial range and variance were 54.4 km (95% CrI: 24.4, 87.7) and 0.33 (95% CrI: 0.21, 0.45), respectively (Table 2).

### 3.4. Model Validation

In children aged 0–23 months, model validation check using 25% of randomly selected locations showed that the predicted model had root mean square error (RMSE) of 18.6, and 57.7% of the predicted proportions were found within 95% CrIs of the posterior predicted distribution. Pearson’s correlation coefficient (r = 0.61) indicated a stronger correlation between observed and predicted values (Appendix A). 

## 4. Discussion

This study showed wide variations in the prevalence of stunting across the administrative zones of Ethiopia. The prevalence of stunting was highest in the South-East zone and the Mekelle Special zone of the Tigray region, while the lowest prevalence was reported in the Sheka zone and the Etang Special zone of the SNNP region. The factors associated with stunting also varied slightly by age group. For children aged 0–23 months, limiting factors were non-receipt of breastmilk, mother’s BMI (overweight/obese), employment status (employed) and higher household wealth. Enabling factors for stunting in children aged 0–23 months included breastfeeding for more than 12 months, residence in the “arid” geographic areas, mother’s perceived birth size of the child (smaller than average) and mother’s BMI (underweight). Almost similar limiting and enabling factors were found among children aged 24–59 months, but with the exception of anemic children, who had a higher likelihood of being stunted compared to that of non-anemic children.

Understanding the specific geographical differences in stunting has several advantages for public health interventions and research in Ethiopia. Firstly, it helps to highlight where new and/or additional public health efforts are needed to tackle childhood undernutrition. Secondly, it helps to ensure that scarce resources are specifically used in regions/areas with the highest burden of the disease. Thirdly, it helps to unmask important geographical heterogeneity and to facilitate subnational comparisons of childhood undernutrition, as country-level estimates cannot provide detailed subnational variations. Finally, it helps to provide more granular subpopulation data, which are essential to the emerging concept of precision public health which has been briefly described as “the use of best available data to target more effectively and efficiently interventions of all kinds to those most in need” [66]. Our study provides detailed subpopulation data on stunting in Ethiopia, where nutrition efforts can be specifically implemented to further reduce the burden of childhood stunting.

Several studies from LMICs have shown clustering of undernutrition within-country regions [67,68,69]. The present study showed that there was a higher proportion of stunted children in the northern Ethiopian regions of Tigray, Afar, Amhara and Benishangul Gumz, and the causes may be multifactorial. The northern regions experience higher than normal natural and manmade shocks, including cyclical drought and famines, civil conflicts and insurgencies [70,71]. These events have important implications for low agricultural production, food insecurity and childhood undernutrition. Additionally, long-lasting high population pressures and deforestation, as well as a high variability in rainfall in the region are likely to have affected land preservation and suitability for crop production and animal grazing [5,71]. Geographically targeted nutritional interventions have the potential to accelerate reductions in childhood stunting through improvement and optimization of resource allocation for programs and services.

The study showed that children who breastfed for more than 12 months were more likely to be stunted compared to those who ceased breastfeeding within 12 months. Studies conducted in Nigeria [46], Nepal [72] and Thailand [73] showed a similar association, where a longer duration of breastfeeding was associated with stunting. WHO/UNICEF recommends EBF from aged 0–5 months and the introduction of timely, diversified, frequent and safe complementary foods to children around the age of six months [74]. However, in many LMICs, the practices of EBF and the timely introduction of complementary foods are often not achievable [75,76,77,78,79,80]. A recent study conducted in India [45] reported that inappropriate complementary feeding was associated with stunting and severe stunting, where unsafe food handling and storage were indicated as one of the enabling factors. These inappropriate food handling practices are evident in Ethiopia [81,82] and may be contributing to the burden of stunting in the country [45].

Consistent with studies from Tanzania [83], Burundi [84], Nigeria [46], India [45] and Nepal [72], our study showed that children perceived by their mothers to be smaller than average size at birth had higher odds of stunting. In the absence of measured birth weight data, mother-reported perceived birth size data have been used as a proxy indicator to approximate birthweight [45,46,72,83,85]. The relationship between smaller birth size and stunting could be due to lower sized children at birth having an increased vulnerability to infection (such as diarrhea, ARI, and malaria) [86,87,88] with resultant complications that include respiratory distress, jaundice, anemia, fatigue and loss of appetite [89,90]. These findings have been reported in research conducted in Nigeria [46], Bangladesh [91] and other LMICs [92]. Mechanisms as to why and how infections increase the risk of childhood stunting have been reported elsewhere [45]. Comprehensive interventions to improving women’s nutritional status, and increasing access and quality of women’s perinatal health services might be beneficial for reducing the burden of stunting attributed to smaller birth size.

Research on the intergenerational effects of childhood undernutrition indicated that perinatal maternal nutritional disadvantage has adverse effects on the health and development of infants and young children [93,94]. These studies showed that children whose mothers were underweight are more likely to be underweight, stunted and wasted. The children also performed worse at school, earned lower income and had a higher risk of non-communicable diseases in adulthood compared to their counterparts [8,94,95]. In the present study, children whose mothers were underweight (i.e., BMI < 18.5 kg/m^2^) had a higher risk of being stunted, but children of mothers who were overweight or obese were less likely to be stunted compared to their counterparts. Similar findings have been reported in studies conducted in Tanzania [83], Nigeria [46], and Pakistan [96]. Maternal underweight possibly contributes to childhood stunting through mother–baby shared genetic factors and the socioeconomic, health and the environmental context in which both mother and child live, and increased the risk of preterm birth and/or LBW from maternal underweight [93,97,98].

Improved household socioeconomic conditions can influence child nutrition through: (i) higher household income, (ii) improved household purchasing power for foodstuffs, and (iii) improved knowledge and childcare practices [1,99,100,101]. This study indicates that children who resided in socioeconomically improved households (i.e., wealthy households or having formally employed mothers/caregivers) were less likely to be stunted. The association between wealthy households and lower odds of stunting was reported in Kenya [102], South Africa [103], and sub-Saharan region [99]. Studies conducted in Uganda [104], South Africa [103], India [45], and sub-Saharan region [81,82,99] also showed the negative influence of mother’s not being employed on childhood stunting.

Our study showed that children from the “arid” geographic areas were associated with stunting, consistent with studies conducted in Uganda [105], Mali [58], and India [49]. The evidence from this study supports the hypothesis of the direct relationship between high aridity index (characterized by excessive heat, and inadequate and variable precipitation) and stunting [52,106]. In Ethiopia, these “arid” geographic areas are associated with stunting potentially due to the impacts of climate change such as frequent and severe shortfalls in precipitation, and continuous rises in temperature, which may result in food insecurity, droughts and undernutrition (including stunting) [107]. Furthermore, more than three quarters of Ethiopians depend on subsistence and rain-fed farming, and livestock production that is historically linked to low crop production, and less diversified and commercial foods [105]. Although this study showed the positive relationship between the “arid” locations and stunting, there is a need for further research in order to examine the mediating effect of crop production and food insecurity with childhood stunting.

## 5. Strengths and Limitations

The use of EDHS data has limitations. Firstly, recall bias due to the interviewer-administered nature of some of the questions may have affected the study results, but objective anthropometric measurement (e.g., height) was used to calculate the nutritional index (HAZ) for stunting [17]. Secondly, the inability to consider all potential confounders (such as food insecurity, and social network factors) may have influenced the measures of association between the study variables and stunting. Nevertheless, climatic factors (e.g., aridity and temperature) were considered in our models, and can approximate factors such as crop production and food insecurity [49,50,51,52]. Thirdly, observed differences in the prevalence of stunting are likely to be under-estimated due to non-differential misclassification bias related to the displacement of GPS coordinates of EDHS clusters, though a circular buffer was drawn for each cluster during the extraction of the geo-covariates [40]. Fourthly, the number of clusters sampled (645 EAs) for the 2016 EDHS was limited, leading to generalizing estimates to the whole of Ethiopia (i.e., 84,915 EAs), nevertheless we implemented a geostatistical model that allowed measurements at any location in the country. Finally, clearly articulating temporal relationships between study factors and outcomes is difficult due to the cross-sectional nature of the study. However, an observational study might be the only method available to investigate some of the study variables (e.g., climatic factors). 

The strengths of the present study include: (i) the nationally representative DHS data; (ii) the use of Bayesian inference, which is superior in modelling geographical dependence of outcomes [65]; (iii) the inclusion of environmental and climatic factors (in addition to the individual level data) may further improve the robustness of our estimates; (iv) the production of subnational surface area maps and with small area stunting prevalence, which can inform resource allocations and implementation of programs in specific areas. 

## 6. Conclusions

Subnational estimates from the administrative zones showed wide variations in the prevalence of stunting in Ethiopia, highlighting the wide heterogeneity in socioeconomic, cultural and climatic risk factors, as well as differences in vulnerability to man-made and natural disasters in Ethiopia. The subpopulation data provides information where nutritional efforts are implemented to further reduce the burden of childhood stunting. Integrated governmental and non-governmental efforts are also needed to break the intergenerational complex interplay of socioeconomic, health, environmental and political factors for childhood stunting. Further research is recommended, to examine whether crop production and food insecurity have mediated the relationship between climatic condition and undernutrition in Ethiopia.

## Figures and Tables

**Figure 1 nutrients-13-02104-f001:**
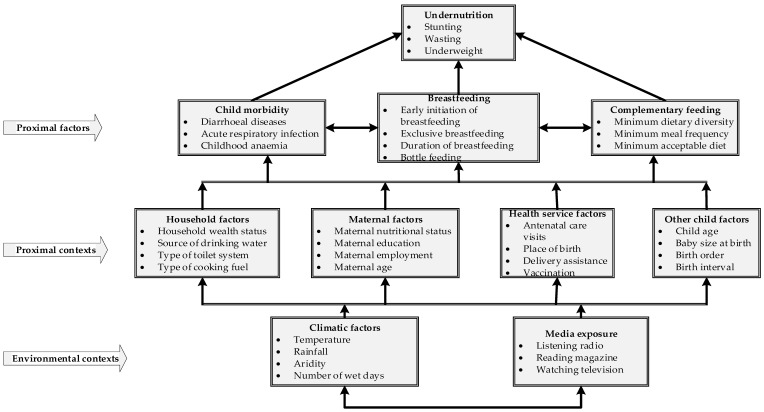
Conceptual framework for proximal and contextual factors associated with stunting among children under five years of age [7,47].

**Figure 2 nutrients-13-02104-f002:**
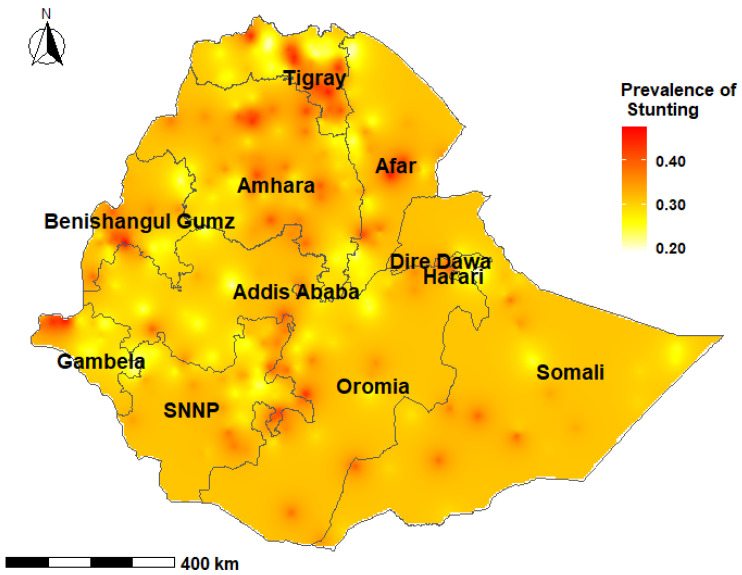
Predicted prevalence of stunting among children 0–23 months of age in Ethiopia, 2016 EDHS.

**Figure 3 nutrients-13-02104-f003:**
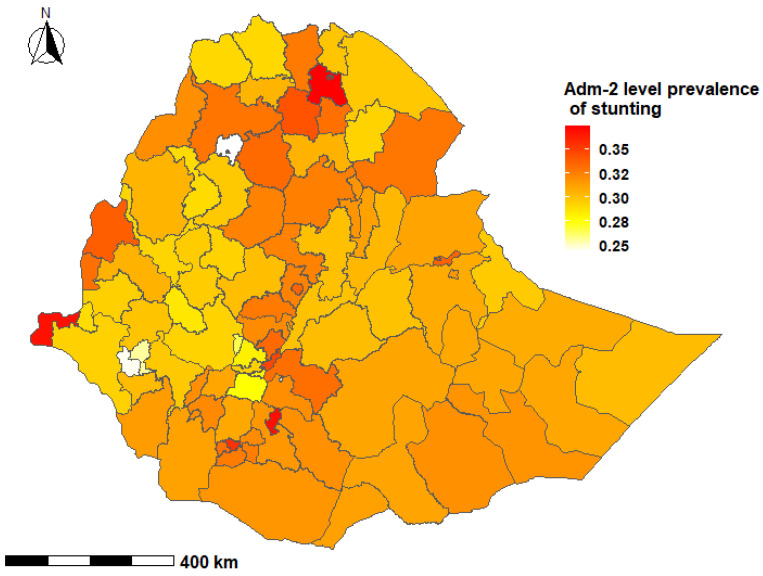
Second administrative level prevalence of stunting among children 0–23 months of age in Ethiopia, EDHS 2016.

**Figure 4 nutrients-13-02104-f004:**
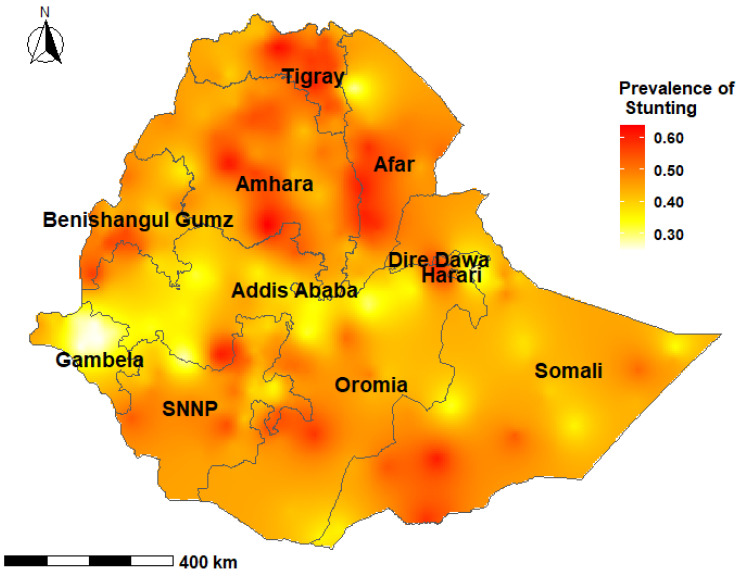
Predicted prevalence of stunting among children 24–59 months of age in Ethiopia, 2016 EDHS.

**Figure 5 nutrients-13-02104-f005:**
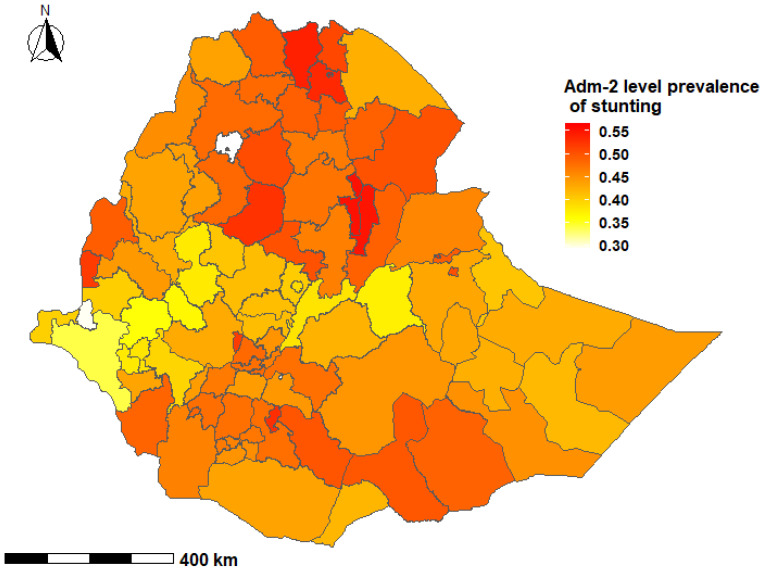
Second administrative level prevalence of stunting among children 24–59 months of age in Ethiopia, EDHS 2016.

**Table 1 nutrients-13-02104-t001:** Non-spatial modelling for proximal and contextual determinants of stunting among children under five years of age in Ethiopia, 2016 EDHS (*n* = 9089).

Variables	0–23 Months of Age	24–59 Months of Age	*p* Interaction for Age
	OR (95% Crl)	OR (95% Crl)	
**Child feeding factors**			
Early initiation of breastfeeding (EIBF)			
No	1.00	-	-
Yes	0.93 (0.77, 1.12)	-	-
Minimum dietary diversity (MDD)			
No	1.00	-	-
Yes	0.86 (0.57, 1.30)	-	-
Minimum meal frequency (MMF)			
No	1.00	-	-
Yes	1.02 (0.83, 1.25)	-	-
Bottle feeding			
No	1.00	-	-
Yes	0.89 (0.70, 1.13)	-	-
Duration of breastfeeding			
≤12 months	1.00	-	-
>12 months	2.03 (1.36, 3.06)	-	-
Overall feeding status (in 24 h)			0.275
Only breastmilk	1.00	1.00	
Breastmilk + supplements	0.87 (0.63, 1.19)	0.50 (0.23, 1.10)	
No breastmilk	0.60 (0.39, 0.90)	0.56 (0.47, 0.67)	
**Other child factors**			
Mother’s perceived baby size at birth			0.005
Larger than average	1.00	1.00	
Average	1.15 (0.92, 1.40)	1.23 (1.07, 1.42)	
Smaller than average	1.35 (1.08, 1.70)	1.68 (1.43, 1.97)	
Diarrhoeal diseases			0.735
No	1.00	1.00	
Yes	1.25 (0.99, 1.57)	1.14 (0.92, 1.41)	
Acute respiratory infection			0.341
No	1.00	1.00	
Yes	1.02 (0.72, 1.46)	1.12 (0.83, 1.51)	
Childhood anaemia			<0.001
No	1.00	1.00	
Yes	1.18 (0.97, 1.44)	1.72 (1.52, 1.96)	
**Maternal factors**			
Maternal nutritional status			0.298
Normal	1.00	1.00	
Underweight	1.36 (1.12, 1.65)	1.19 (1.03, 1.37)	
Overweight/obesity	0.45 (0.30, 0.66)	0.82 (0.64, 1.03)	
Maternal educational status			0.021
No schooling	1.00	1.00	
Primary education	1.00 (0.79, 1.24)	0.99 (0.84, 1.16)	
Secondary or higher education	0.66 (0.44, 1.03)	0.83 (0.60, 1.14)	
Maternal employment status			0.763
No employment	1.00	1.00	
Formal employment	0.70 (0.52, 0.92)	0.95 (0.79, 1.14)	
Informal employment	1.11 (0.90, 1.36)	1.07 (0.92, 1.25)	
**Health service factors**			
Antenatal care visits			0.300
None	1.00	1.00	
1–3 visits	1.16 (0.93, 1.44)	0.87 (0.74, 1.02)	
+4 visits	0.92 (0.72, 1.17)	0.95 (0.81, 1.12)	
Place of birth			0.652
Home	1.00	1.00	
Health facility	0.94 (0.75, 1.17)	1.13 (0.95, 1.34)	
**Household factors**			
Household wealth status			0.001
Poor	1.00	1.00	
Middle	0.68 (0.53, 0.88)	0.89 (0.74, 1.07)	
Rich	0.80 (0.62, 1.03)	0.70 (0.58, 0.85)	
Source of drinking water			0.157
Not protected	1.00	1.00	
Protected	1.03 (0.85, 1.24)	1.05 (0.91, 1.21)	
Toilet system			0.023
Not improved	1.00	1.00	
Improved	0.75 (0.55, 1.01)	0.93 (0.75, 1.15)	
**Climatic factors**			
Daytime land surface temperature			
<30 ° C	1.00	1.00	0.784
30–34.99 °C	0.94 (0.71, 1.22)	1.15 (0.91, 1.46)	
+35 °C	0.99 (0.69, 1.43)	1.13 (0.82, 1.56)	
Annual average rainfall (in mm)			0.863
<141 mm	1.00	1.00	
142–1199 mm	0.53 (0.25, 1.14)	1.06 (0.50, 2.25)	
≥1200 mm	0.46 (0.21, 1.06)	0.96 (0.43, 2.12)	
Aridity			0.138
Wet	1.00	1.00	
Semi-arid	1.67 (1.11, 2.49)	1.32 (0.96, 1.81)	
Arid	2.21 (1.22, 4.02)	2.40 (1.47, 3.93)	
Number of wet days per year			0.071
Low	1.00	1.00	
Medium	1.01 (0.68, 1.49)	0.94 (0.65, 1.36)	
High	1.33 (0.82, 2.14)	1.58 (1.02, 2.46)	
**In-sample model validation**			
DIC	3719.0	6851.9	
WAIC	3719.7	6853.5	
Marginal likelihood	−2108.3	−3687.1	

OR = Odds Ratio; 95% Crl = 95% Credible Interval; DIC = Deviance Information Criterion; WAIC = Watanabe-Akaike Information Criterion.

**Table 2 nutrients-13-02104-t002:** Geospatial modelling for proximal and contextual determinants of stunting among children under five years of age in Ethiopia, 2016 EDHS (*n* = 9089).

Variables	0–23 Months of Age	24–59 Months of Age
	OR (95% Crl)	OR (95% Crl)
**Child feeding factors**		
Early initiation of breastfeeding (EIBF)		
No	1.00	-
Yes	0.91 (0.76, 1.10)	-
Minimum dietary diversity (MDD)		
No	1.00	-
Yes	0.86 (0.57, 1.30)	-
Minimum meal frequency (MMF)		
No	1.00	-
Yes	1.02 (0.83, 1.25)	-
Bottle feeding		
No	1.00	-
Yes	0.89 (0.69, 1.13)	-
Duration of breastfeeding		
≤12 months	1.00	-
>12 months	2.03 (1.36, 3.05)	-
Overall feeding status (in 24 h)		
Only breastmilk	1.00	1.00
Breastmilk + supplements	0.87 (0.63, 1.19)	0.48 (0.22, 1.05)
No breastmilk	0.59 (0.39, 0.90)	0.57 (0.48, 0.67)
**Other child factors**		
Mother’s perceived baby size at birth		
Larger than average	1.00	1.00
Average	1.12 (0.91, 1.38)	1.21 (1.05, 1.39)
Smaller than average	1.35 (1.08, 1.70)	1.64 (1.39, 1.92)
Diarrhoeal diseases		
No	1.00	1.00
Yes	1.28 (1.02, 1.60)	1.17 (0.95, 1.45)
Acute respiratory infection		
No	1.00	1.00
Yes	1.05 (0.74, 1.48)	1.10 (0.82, 1.48)
Childhood anaemia		
No	1.00	1.00
Yes	1.17 (0.96, 1.43)	1.73 (1.52, 1.96)
**Maternal factors**		
Maternal nutritional status		
Normal	1.00	1.00
Underweight	1.36 (1.11, 1.65)	1.21 (1.05, 1.40)
Overweight/ obesity	0.47 (0.31, 0.69)	0.84 (0.66, 1.06)
Maternal educational status		
No schooling	1.00	1.00
Primary education	0.98 (0.78, 1.24)	0.99 (0.84, 1.17)
Secondary or higher education	0.67 (0.44, 1.01)	0.86 (0.63, 1.18)
Maternal employment status		
No employment	1.00	1.00
Formal employment	0.68 (0.51, 0.91)	0.95 (0.79, 1.14)
Informal employment	1.05 (0.85, 1.30)	1.00 (0.86, 1.17)
**Health service factors**		
Antenatal care visits		
None	1.00	1.00
1–3 visits	1.16 (0.93, 1.45)	0.86 (0.73, 1.00)
+4 visits	0.91 (0.71, 1.16)	0.94 (0.80, 1.11)
Place of birth		
Home	1.00	1.00
Health facility	0.93 (0.75, 1.16)	1.12 (0.94, 1.33)
**Household factors**		
Household wealth status		
Poor	1.00	1.00
Middle	0.68 (0.53, 0.89)	0.90 (0.74, 1.08)
Rich	0.80 (0.62, 1.04)	0.71 (0.59, 0.85)
Source of drinking water		
Not protected	1.00	1.00
Protected	1.02 (0.84, 1.24)	1.03 (0.90, 1.19)
Toilet system		
Not improved	1.00	1.00
Improved	0.74 (0.54, 1.01)	0.96 (0.77, 1.19)
**Climatic factors**		
Daytime land surface temperature		
<30 °C	1.00	1.00
30–34.99 °C	0.91 (0.67, 1.22)	1.12 (0.87, 1.45)
+35 °C	1.01 (0.66, 1.54)	1.19 (0.82, 1.73)
Annual average rainfall (in mm)		
<141 mm	1.00	1.00
142–1199 mm	0.53 (0.21, 1.41)	0.91 (0.36, 2.31)
≥1200 mm	0.52 (0.19, 1.49)	0.85 (0.32, 2.26)
Aridity		
Wet	1.00	1.00
Semi-arid	1.67 (1.11, 2.49)	1.33 (0.93, 1.91)
Arid	2.21 (1.22, 4.02)	2.02 (1.11, 3.65)
Number of wet days per year		
Low	1.00	1.00
Medium	0.96 (0.59, 1.54)	1.25 (0.79, 1.98)
High	1.20 (0.67, 2.12)	1.77 (1.05, 2.99)
**In-sample model validation**		
DIC	3687.5	6845.6
WAIC	3691.5	6848.0
Marginal likelihood	−2115.9	−3681.5
**Spatial random effects**		
Kappa	7.33 (2.14, 11.86)	6.43 (2.92, 10.65)
Variance	0.24 (0.10, 0.39)	0.33 (0.21, 0.45)
Range * (in km)	52.2 (15.5, 98.8)	54.4 (24.4, 87.7)

OR = Odds Ratio; 95% Crl = 95% Credible Interval; DIC = Deviance Information Criterion; WAIC = Watanabe-Akaike Information Criterion; * Range indicates the distance value (in the unit of the point coordinates) above which spatial dependencies become negligible.

## Data Availability

Datasets are available to at https://www.dhsprogram.com/data/ (accessed on 15 November 2019).

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
