# Peer review of "Mapping Geographical Differences and Examining the Determinants of Childhood Stunting in Ethiopia: A Bayesian Geostatistical Analysis"

_nutrients, 2021, doi:10.3390/nu13062104_

Round 1

Reviewer 1 Report

This study estimated high-spatial-resolution stunting risk among children under five years old in Ethiopia, based on 2016 Ethiopia DHS data, potential influencing predictors and Bayesian geostatistical modeling. The logic of the study is clear and the conceptual framework for potential predictors is reasonable. The risk maps either at high-resolution or at district-level can provide valuable support for spatial-targeted control strategies for child stunting in Ethiopia. However, there are several points needed to clarify as following:

1) The authors categorized children into two age groups: 1-23 months and 24-59 months. Please clarify the reasons to make this cut-off. And biological or pathophysiological considerations?

2) The authors analyzed data separately for age groups 0-23 months, 24-59 months and 0-59 months. In fact, if the authors assumed that groups 0-23 months and group 24-59 months were inhomogeneous and may have different influencing factors, what are the reasons to combine the two groups and did additional analysis for all children, without adding age and age-factor interaction terms in the model? If the authors wanted to produce a risk map of children for overall ages, then combining the two risk maps of 0-23 months and 24-59 months with weighting for population of the two groups is enough to produce such map.

3) The authors transformed all continuous variables to categorical ones in the model. Why didn’t them consider the original forms of variables as independent variables in the model? The transformation may lead to loss of information. Please clarify it.

4) As the number of potential predictors is large, did the authors consider variable selection for the best set of predictors in the final geostatsitcal model? According to Table 1, sees all variables were included in the final model. For a parsimonious model, variable selection is necessary.

5) To produce high-resolution maps, data of predictors at high-resolution is required. However, as many of the factors, such as duration of breastfeeding, diarrheal diseases, childhood anemia, were obtained from participants in DHS available for each survey cluster, how to assign those data to the whole study surface when estimate the risk of the outcome variable?

6) Other model validation indicators, such as mean error, mean absolute error, AUC and coverage of observations within 95% BCI of the posterior predictions, should be reported for better assessment of the model performance.

7) The authors should provide more details about how the geographical models were constructed, including the assumption of the data distribution (binomial? Bernoulli? Gaussian?) and the type of the link function (logistic? linear?).

8) The authors just mentioned that they used non-informative priors, but what is the exact prior distribution for each parameter/hyper-parameter? Please list them in the manuscript, as setting different priors, even they are non-informative, may influence the final posteriors.

9) The authors may miss to list the detailed source of demographical data. Please add it.

Reviewer 2 Report

Attached comments.
